# Increased Immunogenicity of Full-Length Protein Antigens through Sortase-Mediated Coupling on the PapMV Vaccine Platform

**DOI:** 10.3390/vaccines7020049

**Published:** 2019-06-12

**Authors:** Marie-Ève Laliberté-Gagné, Marilène Bolduc, Ariane Thérien, Caroline Garneau, Philippe Casault, Pierre Savard, Jérome Estaquier, Denis Leclerc

**Affiliations:** 1Department of Microbiology, Infectiology and Immunology, faculty of Medicine, Laval University, Quebec City, QC G1V 4G2, Canada; marie-eve.L-Gagne@crchudequebec.ulaval.ca (M.-È.L.-G.); marilene.bolduc@crchudequebec.ulaval.ca (M.B.); ariane.therien@crchudequebec.ulaval.ca (A.T.); caroline.garneau@crchudequebec.ulaval.ca (C.G.); philippe.casault@crchudequebec.ulaval.ca (P.C.); jerome.estaquier@crchudequebec.ulaval.ca (J.E.); 2Department of Neurosciences, faculty of Medicine, Laval University, Quebec City, QC G1V 4G2, Canada; pierre.savard@crchudequebec.ulaval.ca

**Keywords:** vaccine platform, nanoparticle, papaya mosaic virus

## Abstract

*Background*: Flexuous rod-shape nanoparticles—made of the coat protein of papaya mosaic virus (PapMV)—provide a promising vaccine platform for the presentation of viral antigens to immune cells. The PapMV nanoparticles can be combined with viral antigens or covalently linked to them. The coupling to PapMV was shown to improve the immune response triggered against peptide antigens (<39 amino acids) but it remains to be tested if large proteins can be coupled to this platform and if the coupling will lead to an immune response improvement. *Methods*: Two full-length recombinant viral proteins, the influenza nucleoprotein (NP) and the simian immunodeficiency virus group-specific protein antigen (GAG) were coupled to PapMV nanoparticles using sortase A. Mice were immunized with the nanoparticles coupled to the antigens and the immune response directed to the antigens were analyzed by ELISA and ELISPOT. *Results*: We showed the feasibility of coupling two different full-length proteins (GAG and NP) to the nanoparticle. We also showed that the coupling to PapMV nanoparticles improved significantly the humoral and the cytotoxic T lymphocyte (CTL) immune response to the antigens. *Conclusion*: This proof of concept demonstrates the versatility and the efficacy of the PapMV vaccine platform in the design of vaccines against viral diseases.

## 1. Introduction

Recent advances in vaccinology lead to the development of subunit vaccines, in replacement of the traditional live attenuated or inactivated vaccines, to ease and accelerate manufacturing and to maximize safety. One drawback of this strategy is the weak immunogenicity of the subunit vaccines and the multiple injections necessary for reaching protective antibody titers. To alleviate this problem, the multimerization of recombinant antigens onto nanoparticles was shown to increase both antigen immunogenicity and stability [1,2,3,4]. This approach was successful with the human papillomavirus (HPV) vaccine composed of the recombinant L1 major coat-protein self-assembled into empty shells mimicking the HPV structure [5]. Efficacy of the HPV vaccine is linked to its highly immunogenic repetitive structure that is recognized by dendritic cells, therefore, leading to a robust immune response that generates protective levels of HPV specific neutralizing antibodies [6]. The multimerization of the HPV coat protein (CP) is straightforward since it is its main function to self-assembled into virus-like particles (VLPs). To profit of the advantage of the multimerization, recombinant proteins that are unable to self-assemble must be linked to a highly ordered structure like a VLP or a nanoparticle [7].

The self-assembly of the papaya mosaic virus coat protein (CP) onto an single-stranded RNA (ssRNA) leads to the formation of a flexuous rod-shaped nanoparticle that triggers innate immunity [8] through the stimulation of toll like receptors (TLRs)7/8 [2,9]. The PapMV nanoparticle has been shown to be an efficient platform for the presentation of peptide antigens to the immune system [7,10,11,12]. For short peptides, the DNA sequence encoding the antigen is fused to the PapMV coat protein gene. Therefore, each recombinant coat protein is decorated with an antigenic peptide and the resulting nanoparticle is presenting hundreds of copies of the peptide to the immune system [4,10,11,12,13]. However, this approach is limited to short peptides because the size and the nature of the fused antigen can interfere with the CP self-assembly, an essential prerequisite for immunogenicity enhancement [11]. To overcome this limitation, we have been using a bacterial transpeptidase (sortase A, SrtA) to directly attach the antigen onto PapMV nanoparticles [7].

SrtA is an enzyme found on the bacterial surface that reacts with proteins, such as virulence factors or cell wall peptidoglycans through the introduction of a covalent link. Target protein A, which possess the (LPTXG) recognition motif, gets cleaved by SrtA leading to the presentation of a thioester acyl-enzyme intermediate. The attack of this intermediate by the amine nucleophile of the poly-glycines (G) (usually 3 to 5 G) of protein B covalently linked the two partners and regenerate a free SrtA [14,15,16]. 

Using this approach, large peptides ranging from 26 (influenza M2 peptide) to 39 (HIV T20 peptide) amino acids were covalently coupled to PapMV nanoparticles. Immunization with these vaccines was shown to protect mice from an influenza virus challenge [7].

Aim: These results are very promising, but to extend the potential of this approach, it remains to be determined if a full-length protein antigen can be coupled onto the PapMV nanoparticle and if the coupling of large antigen to the vaccine platform can improve its immunogenicity.

Conclusion: In this brief report, we have made a strong proof of concept demonstrating the feasibility of coupling large full-length proteins to the PapMV vaccine platform using sortase A. We also showed that coupling to the nanoparticle significantly enhanced the antibody (humoral) and the cellular cytotoxic T lymphocyte (CTL) immune responses directed toward the coupled large protein antigens, showing the potential of this technology in the design of vaccines to viral diseases.

## 2. Materials and Methods

### 2.1. Production of the Recombinant Proteins

Production methods for both PapMV nanoparticles and sortase A have been previously reported [7]. Two large protein antigens, one derived from the group-specific antigen (GAG) protein of the Simian immunodeficiency virus (SIV) and the other derived from two conserved proteins of the influenza virus (NP and M2e), were produced using recombinant DNA technology.

The GAG antigen is a truncated portion of the protein (MA_1–15_-CA-NC_100–447_), where the membrane-anchoring domain of MA (matrix) has been deleted to increase protein solubility (Figure 1A). The recombinant GAG gene has been fused serially at its N-terminus to a 6xH tag, the tobacco etch (TEV) protease specific cleavage site, and three glycines. The recombinant protein was expressed in *Escherichia coli* (BD792) and purified using immobilized-metal-ion-chromatography (IMAC). Treatment of the purified recombinant GAG with the tobacco etch virus (TEV) protease (Genescript, Piscataway, NJ, USA) added glycine to the GGGG motif and released GAG (40 kDa) from its 6xH tag.

The second recombinant protein resulted from the serial fusion of the glutathione S-transferase (GST) gene, the TEV protease cleavage site, four glycines, the influenza NP gene, the influenza sM2e sequence encoding a short peptide antigen, and finally, a 6xH tag at the C-terminus. The recombinant protein (85 kDa) was expressed in *E. coli* (BL21) and purified using IMAC. Treatment of the purified recombinant protein with the TEV protease added glycine to the GGG motif and released the recombinant NP-M2e-His-Tag (58 kDa); the contaminating free GST was removed by chromatography on glutathione immobilized agarose beads. A clone without the sM2e peptide was produced to perform the ELISA anti-NP. The production and the purification process was identical to the NP-sM2e clone.

### 2.2. Coupling Reactions with SrtA

The coupling of the recombinant GAG and NP-M2e antigens onto the PapMV nanoparticles was conducted according to Thérien et al. 2017 [7]. In brief, several coupling reactions using various ratios of PapMV nanoparticles versus recombinant proteins were conducted to select the most efficient reaction. The optimal ratio of PapMV CP:sortase:GAG (in µM) was 80:4:5, whereas it was 10:2:2 for PapMV CP:sortase:NP-M2e. The coupling reactions were incubated for 2.5 h at 22 ± 2 °C and stopped using 10 mM EDTA. The final coupled nanoparticles were purified by size-exclusion chromatography on a Superdex 200 column (GE Healthcare, Baie d’Urfe, Canada).

### 2.3. SDS-PAGE for Coupling Efficacy and Western Blot

Samples were denatured by heating at 95 °C for 10 min in a solution containing a dye and 2-mercaptoethanol. Then, samples were loaded on 15% Tris-Glycine sodium dodecyl sulfate polyacrylamide gel electrophoresis (SDS-PAGE). To calculate the coupling efficacy, we measured the intensity of the coupled CP (higher molecular weight on the gel; Figure 1B,D) on the total CP (coupled plus uncoupled). The intensity of the signals was estimated using the ImageJ software (ImageJ 1.52a). The identity of the bands visualized on the gel was assessed by Western blotting using anti-SIVmac p27 Monoclonal (55-2F12) (NIH, Bethesda, Maryland, USA), an anti-Influenza A virus NP (Abcam, Toronto, On, Canada) and an in-house made anti-PapMV CP polyclonal antibody.

### 2.4. Immunization and Immune Response

Balb/C mice, 5 per group, were immunized twice by the intramuscular route with the antigen alone (9 µg NP; 9.75 µg GAG), the antigen mixed with the PapMV nanoparticles (9 µg PapMV plus 90 µg NP-M2e; 9.75 µg PapMV plus 55.25 µg GAG), the antigen coupled to the PapMV nanoparticles (90 µg PapMV-NP-M2e; 65 µg PapMV-GAG) and the buffer. Blood was harvested at 13 (NP-M2e) or 20 (GAG) days after the first immunization, and/or at 28 (NP-M2e) or 33 (GAG) days after the second immunizations made at day 14. The sera were used to reveal the presence of antibodies directed to each of the vaccine antigens. ELISA directed to either GAG or NP was performed to assess the humoral response as previously reported [7]. ELISPOT (day 34), where GAG was used to stimulate the splenocytes, was performed to assess the CTL response following a previously reported protocol [10]. For the ELISPOT (day 28) with NP-M2e formulations, the immunodominant TYQRTRALV Balb/C CTL epitope derived from NP-M2e was used to stimulate splenocytes isolated from mice immunized with the NP-M2e antigen as previously reported [13].

### 2.5. Statistics

Data were analyzed with Graph Pad PRISM 7.0 software (GraphPad Software, Inc., San Diego, CA) for statistical significance using the ANOVA test. Tukey’s post-test was also used to compare the difference among groups of mice. Values of *p* smaller than 0.05 were considered significantly different.

### 2.6. Ethics Statement

All animal work was previously approved by the “Comité de Protection des Animaux—CHUQ (CPA-CHUQ)” of the institution. The authorization numbers were 2013142-3 and 2017115-1.

## 3. Results

### 3.1. Coupling of GAG and NP to the PapMV Nanoparticles

The recombinant GAG (Figure 1A), the influenza NP-M2e (Figure 1C) and the PapMV coat protein (CP) (Figure 1B,D) were all produced in an *E. coli* expression system and purified by immobilized-metal-ion-chromatography (IMAC) in native conditions. The PapMV CP was self-assembled in vitro by mixing the CP with an ssRNA of 1517 nucleotides (nt) long, leading to the generation of PapMV nanoparticles (nano) as previously reported [7]. 3-D modeling of PapMV nano [7] predicted the C-terminus of the CP to be located in the interior of the nanoparticle and exposed to each extremity of the nanoparticle. Therefore, the sortase receptor motif (LPETGG), followed by the 6xH tag, was only available for reaction with SrtA on the extremities of the nanoparticle. This arrangement preserves the surface of the nanoparticle, which is probably involved in the immune enhancement properties of the PapMV nanoparticles as previously described [7].

The GAG and NP-M2e recombinant proteins will be cleaved with the TEV protease to present at their N-terminus the free glycine (G) residues, corresponding to the sortase donor motif. The coupling reactions were performed by mixing the PapMV nanoparticles with GAG or NP-M2e in presence of the SrtA to induce the formation of a covalent link between the nanoparticles and the antigens. The proportion of nanoparticles coupled to GAG was estimated to be 10–15% (Figure 1B; left panel, lane 4). A similar coupling efficacy was obtained with the NP-M2e antigen (Figure 1D, left panel, lane 4). With the GAG antigen, coupling reactions exceeding 15% efficacy were obtained, but they resulted in the formation of aggregates that were not suitable for vaccination. Almost all the antigens added to the coupling reaction (Figure 1B; left panel lane 3 and Figure 1D; left panel lane 3) were coupled to the PapMV CP with a covalent link, leading to a shift of the molecular weight of the fusion product (Figure 1B; left panel lane 4 and Figure 1D; left panel, lane 4). Western blot analysis also revealed the fusion products using the anti-GAG, the anti-NP and the anti-PapMV CP-specific antibodies, which showed, respectively, a molecular weight of 64 kDa for the fusion of GAG and PapMV CP (Figure 1B; middle and right panels, lane 3) and 82 kDa for the fusion of NP and PapMV CP (Figure 1D; middle and right panels, lane 3).

To assess if the fusion of the antigen altered the nanoparticles, the size and the appearance of the free and coupled nanoparticles were analyzed by dynamic light scattering (DLS) and electron microscopy (Appendix A). Both coupled nanoparticles appeared larger in size than the free nanoparticle (Appendix A). It is likely that the coupling of the antigens on the nanoparticle-induced non-specific aggregation, which impacted their apparent size on the DLS. However, the formulations made with the coupled-nanoparticles were stable at 4 °C and did not show any precipitate even several weeks after their synthesis. The electron microscopy images revealed also that the coupled PapMV nanoparticles tended to aggregate in comparison to the free PapMV nanoparticles (Appendix A). This supported our hypothesis that aggregation of the coupled nanoparticles is responsible for their increment in size in comparison to the free PapMV nano.

### 3.2. Assessment of the Immune Response

The immunoglobulins (IgG and IgG2a) titers of immunized animals were assessed after one and two immunizations, by ELISA, directed to the GAG (Figure 2) or NP antigens (Figure 3). The levels of total IgG of the animals immunized with the coupled GAG antigen (Nano-GAG), the uncoupled GAG mixed with PapMV nano (nano plus GAG) or GAG alone were comparable after a single immunization (Figure 2A). However, only the mice immunized with PapMV and GAG (coupled or not) showed IgG2a titers that significantly exceeded those of the mice immunized with GAG alone (Figure 2B). After two immunizations, the humoral response (total IgG and IgG2a) was increased in all groups. The group immunized with nano-GAG showed significantly higher total IgG titers when compared to the group immunized with GAG alone (Figure 2C). When the IgG2a titers were assessed, both groups containing the PapMV nano (coupled and not coupled) showed a significant increase when compared to the group immunized with GAG alone (Figure 2D). The adjuvant property of the PapMV nanoparticles was probably responsible for the increase of titers in the nano plus GAG group [10,11,17,18,19]. However, the highest antibody titers to GAG were recorded in the mice immunized with coupled nano-GAG (Figure 2C,D). The coupling of the antigen to the PapMV nano (nano-GAG) significantly enhanced the capacity of CD4^+^ and CD8^+^ T cells of the vaccinated mice to produce interferon-γ (IFN-γ) upon stimulation of the splenocytes with the purified GAG antigen when compared with the antigen alone (Figure 2E). Interestingly, the nano plus GAG formulation did not significantly increase the IFN-γ secreting cells when compared with the antigen alone (Figure 2E).

Mice immunized with the NP antigen coupled to PapMV nano (nano-NP) showed a significant increase of their total IgG and IgG2a titers as compared to the uncoupled formulation (nano plus NP) or the NP alone (Figure 3A,B). The significant difference, however, between the coupled and the uncoupled group was lost after two immunizations (Figure 3C,D) but the highest antibody titers (total IgG and IgG2a) were obtained in mice immunized with the coupled vaccine. The animals vaccinated with the coupled and uncoupled antigen revealed significantly higher IgG and IgG2a titers when compared to the NP alone (Figure 3C,D). We have not been able to reveal a significant amount of antibodies directed to the M2e peptide, which is fused to the C-terminus of the NP antigen (Appendix A). Masking of the M2e peptide by the NP protein may explain the lack of B cell response to this epitope. Finally, only mice vaccinated with the coupled vaccine showed a significant improvement of the CTL response by ELISPOT as compared to the other groups, confirming for the second time the capacity of the vaccine platform to enhance the T cell immune response (Figure 3E). In this experiment, the isolated splenocytes were stimulated with an immunodominant Balb/C CTL epitope, which suggests that the platform was effective for the induction of the proliferation of CD8^+^ T cells specific to this NP epitope.

## 4. Discussion

In this study, we have demonstrated that the coupling of full-length viral proteins to PapMV nanoparticles (vaccine platform) using a SrtA coupling approach led to a significant improvement of immunogenicity. The coupling efficacies (10–15%) obtained with a large polypeptide is at least two times lower than those obtained with small peptides (30%) [7]. The steric hindrance of the protein antigen attached to PapMV nano probably masks other available SrtA recognition motifs and reduces the coupling efficacy. Our data also revealed that the coupling of both antigens to the PapMV nanoparticles was critical to elicit an efficient CTL immune response to the antigen. The PapMV nanoparticle is a natural TLR7/8 agonist, which reaches the endosome of immune cells [9]. Upon coupling, the nanoparticle would drag the antigen to the endosomal compartment, where it would get degraded, leading to an improvement of the antigenic presentation of the CTL epitopes of the antigens on the major histocompatibility complex (MHC) class I and II [20]. In addition, the PapMV nanoparticle also carries an ssRNA, which is released in the endosome and stimulates TLR7, leading to the activation of innate immunity, which will contribute to the enhancement of the immune response directed to the antigen [2].

The humoral immune response reached the highest antibody titers when the PapMV nanoparticle was coupled to the antigen. The combination of the PapMV nanoparticle with the antigen, without covalent linking, also showed a significant improvement in antibody titers to the antigens due to the intrinsic adjuvant property of the PapMV nanoparticles [11,17,19,21]. The benefit of the coupling of the antigens on the vaccine platform to trigger the humoral response was more noticeable after a single injection, this observation became less apparent after the boost immunization. The fact that full-length proteins, in contrast to small peptides, are more immunogenic, might also explain why we did not observe major differences between the coupled and the non-coupled formulations after two injections.

## 5. Conclusions

The coupling of full-length proteins to PapMV nanoparticles was made possible using bacterial SrtA. This coupling reaction was simple to perform and insured the attachment of the protein antigen to the vaccine platform through a covalent bond. These results showed that this approach allows the development of vaccine formulation capable to trigger a strong CTL immune response needed to prevent viral diseases or for the development of vaccines in cancer immunotherapy [22,23].

## Figures and Tables

**Figure 1 vaccines-07-00049-f001:**
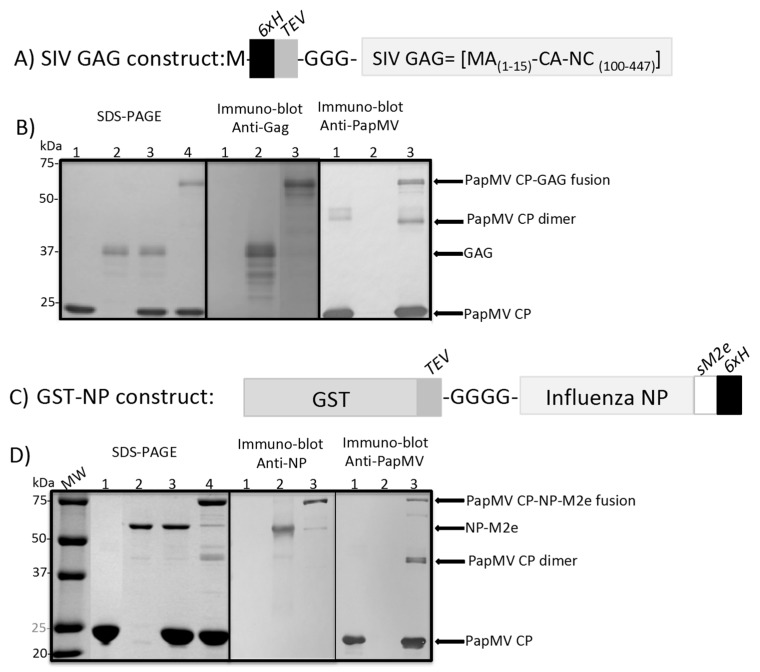
Coupling of the recombinant proteins to the PapMV nanoparticle (nano) vaccine platform using sortase A (SrtA). (**A**) Schematic representation of the simian immunodeficiency virus (SIV) group-specific antigen (GAG) protein clone. In brief, a 6xH tag was fused directly after the first methionine (M) start codon followed by the tobacco etch virus (TEV) protease cleavage site, the GGG SrtA donor motif, the N-terminus of the SIV matrix (MA 1–15) and the fragment 100–447 of the SIV-capsid and nucleocapsid (CA-NC). Upon cleavage with the TEV, an extra G is added to the N-terminus of GAG, which generates the four glycine (GGGG) N-terminus of the protein. (**B**) SDS-PAGE (left panel), 1: PapMV nano (2 µg), 2: SIV GAG (0.3 µg), 3: PapMV nano (2 µg) + SIV GAG (0.3 µg) and 4: Nano-GAG (2 µg). Molecular weights markers are shown to the left of the gel. Immuno-Blot performed with anti-SIV mac p27 (middle panel), 1: PapMV nano (2 µg), 2: SIV GAG (0.3 µg), 3: Nano-GAG (2 µg). Immuno-Blot performed with anti-PapMV-CP (right panel). 1: PapMV nano (1 µg), 2: SIV GAG (0.15 µg), 3: Nano-GAG (1 µg). (**C**) Schematic representation of the influenza nucleoprotein (NP-sM2e) clone. The glutathione S-transferase (GST) was fused at its C-terminus to the TEV protease specific cleavage site followed by GGGG, the influenza NP gene, the influenza sM2e peptide and a 6xH tag. (**D**) SDS-PAGE (left panel), 1: PapMV nano (10 µg), 2: NP (1 µg), 3: PapMV nano (10 µg) + NP (1 µg) and 4: Nano-NP (10 µg). Molecular weights markers are shown to the left of the gel. Immuno-Blot anti-NP (middle panel), anti-PapMV CP (right panel), 1: PapMV nano (1 µg), 2: NP (1 µg), 3: Nano-NP (1 µg).

**Figure 2 vaccines-07-00049-f002:**
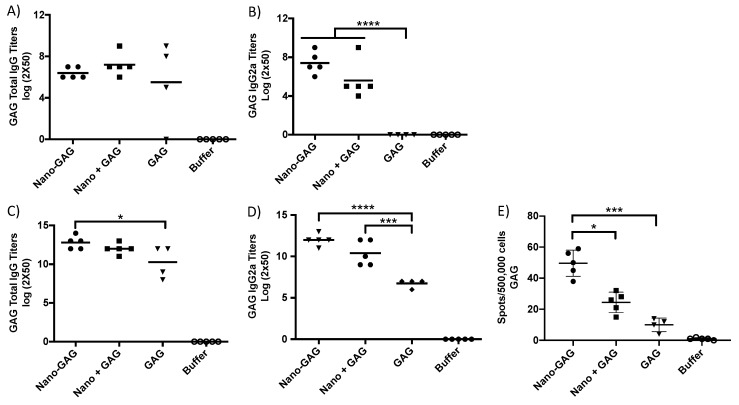
Coupling of the SIV GAG to the PapMV nanoparticle enhances the immune response directed to the GAG antigen. Balb/C mice, 5 per group, were immunized twice by the intramuscular route (i.m.) at a 21 day interval with, respectively, PapMV nano coupled to the Gag antigen (nano-GAG) (65 µg), PapMV nano (55.25 µg) mixed with GAG (9.75 µg) (Nano plus GAG), GAG alone (9.75 µg) and buffer control (10 mM Tris, 150 mM NaCL pH8). The assessment of the humoral response directed to the GAG recombinant antigen was performed with serum harvested at day 20 (**A**,**B**) or at day 33 (**C**,**D**). The anti-GAG total IgG (**A**,**C**) and IgG2a titers (**B**,**D**) were measured. The spleen was harvested at day 34 to perform the ELISPOT assay (**E**). The splenocytes were activated with the SIV GAG recombinant proteins. * *p* < 0.05, *** *p* < 0.001, **** *p* < 0.0001.

**Figure 3 vaccines-07-00049-f003:**
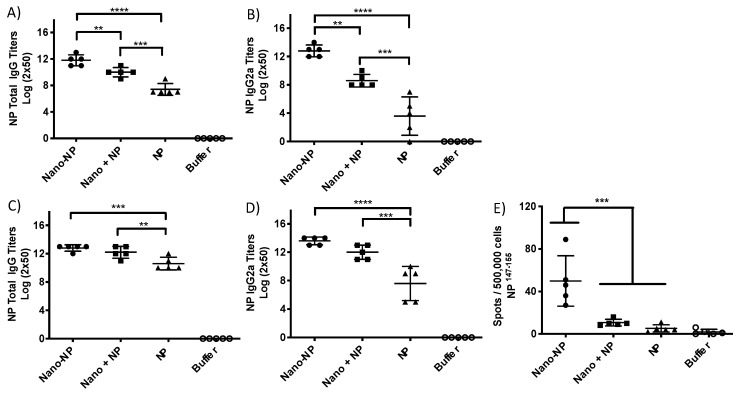
Coupling of NP to PapMV nanoparticle enhances the immune response directed to the NP antigen. Balb/C mice, 5 per group, were immunized twice i.m. at a 14 day interval with PapMV nano coupled to the NP antigen (nano-NP) (90 µg), PapMV nano (90 µg) mixed with NP (9 µg) (Nano plus NP), NP alone (9 µg) and buffer control (PBS 1X). The assessment of the humoral response directed to the NP recombinant antigen was performed with serum harvested at day 13 (**A**,**B**) and at day 28 (**C**,**D**). The anti-NP total IgG (**A,C**) or the anti-NP IgG2a titers (**B**,**D**) were measured. The spleen was harvested at day 28 to perform the ELISPOT assay (**E**). The splenocytes were activated with the CTL epitope NP_147-155_. ** *p* < 0.01, *** *p* < 0.001, **** *p* < 0.0001.

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
