# Peer review of "Increased Immunogenicity of Full-Length Protein Antigens through Sortase-Mediated Coupling on the PapMV Vaccine Platform"

_vaccines, 2019, doi:10.3390/vaccines7020049_

Round 1
Reviewer 1 Report
The paper is very interesting and I like the attention given to the immunological studies. Two comments I would like to make are:
it is unclear from the Westerns on Figure 1B and D, lane 4, whether I am looking at an assembled nanoparticle here. The mw for the top band is close to 75 kD. So this is a fusion protein? How do I know that any of this assembled into a nanoparticle,and how long these particles are (compared to wt)? What is the cause of the difference in assembly ratios between GAG and NP? Size and charge? From what I can see, I have no way of knowing that you have assembled nanoparticles except to look up reference 7.
Any comments regarding the low efficacy of sortase mediated conjugation? Is it just that only a few CP fuse to antigen or is it because of steric hindrance that only 15% of antigen is present in nanoparticle? Do you think it will be possible in the future to increase this percentage, and how would you propose to do that?
Some grammar problems in parts of manuscript: eg. First page, Introduction; line 31 should be "to maximize safety"; line 33 should be "was shown" etc.
Author Response
Reviewer 1
The paper is very interesting and I like the attention given to the immunological studies. Two comments I would like to make are:
(1) it is unclear from the Westerns on Figure 1B and D, lane 4, whether I am looking at an assembled nanoparticle here. The mw for the top band is close to 75 kD. So this is a fusion protein?
Answer : Dear reviewer thank you for your comment. To clarify the manuscript, we have modified the Figure 1B and D and identified with arrows the protein bands found on the gel and the western blots. Respectively, the PapMV CP monomer (Figure 1B and 1D), the NP-M2e (Figure 1B), the GAG (Figure 1D) and both fusion products (Figure 1B and 1D) were pointed with black arrows. We also identified the PapMV CP dimer (Figure 1B and D) that is the result of an incomplete denaturation of the PapMV nanoparticles on the SDS-PAGE.
(2) How do I know that any of this assembled into a nanoparticle and how long these particles are (compared to wt)?
Answer : Dear Reviewer thank you for your comment. We have added in supplementary figure S1A the dynamic ligth scattering (DLS) of the free PapMV nano and those coupled to the NP-M2e or the GAG antigens. At Figure S1B, we also showed the EM images of the free PapMV nano and coupled to either the NP-M2e and the GAG. These data reveal that the coupling of the antigens to the PapMV nanoparticles do not affect the appearance of the nanoparticles. However, the size of the nanoparticles has changed by the coupling. We have added lanes 174-184 to explain these observations. We prefer to keep this information in supplementary data because there are not critical for this manuscript.
(3) What is the cause of the difference in assembly ratios between GAG and NP?
Answer : Dear reviewer, thank you for your question. The coupling reactions for each antigen must be optimised. Those conditions will change depending of the nature of the antigen. Several factors can affect the conditions of coupling including the solubility of the antigen or the coupled antigen-nano complex. For example, the GAG antigen tends to aggregate at high concentration and also, as mentioned in the manuscript (lane 165-166), a higher density of coupling of GAG onto the PapMV nano induced formation of a precipitate that is unwanted for a vaccine preparation. Therefore, we were obliged to work with a ratio of PapMV CP/SrtA/antigen of 80 :4 :5 (µg/mL). This ratio contains less antigen in the coupling reaction than when the NP-M2e antigen is used because NP-M2e is more soluble allowing an optimal ratio of 80 :16 :16 (µg/mL). It also possible that the size of the antigen will have an impact on the total amount that we will be able to couple on the PapMV nano. As mentioned in the discussion (lane 281-283) that we have modified accordingly, the steric hindrance of large protein like the NP-M2e that are attached to the PapMV CP can mask an adjacent SrtA receptor motif and consequently affect the efficacy of coupling. Because peptides are smaller, it is possible to increase the coupling efficacy as compared to the large NP-M2e antigen and as previously reported (Thérien et al., 2017; ref. 7).
(4) Size and charge? From what I can see, I have no way of knowing that you have assembled nanoparticles except to look up reference 7.
Answer : (see response question 2, and the new Figure S1B)
(5) Any comments regarding the low efficacy of sortase mediated conjugation? Is it just that only a few CP fuse to antigen or is it because of steric hindrance that only 15% of antigen is present in nanoparticle?
Answer :(see answer question 3 and change to the discussion lane 281-283)
(6) Do you think it will be possible in the future to increase this percentage, and how would you propose to do that?
Answer : Dear reviewer, thank you for your comment. We are currently developing a second version of the vaccine platform where the antigen will be fused to the surface of the nanoparticle instead of the extremities as it is showed in this manuscript. It is anticipated that more SrtA receptor motif will be available at the surface of the nanoparticle, which will allow increasing potentially the % of efficacy of the coupling. However, these fusions will also mask the surface of the PapMV nanoparticle that migth be important for the recognition of the nanoparticle by the immune cells and for their targeting to the endosome where the antigen processing is performed. We are currently investigating this issues and hopefully we will be able by the end of this year to publish our results on a second generation of the PapMV vaccine platform.
(7) Some grammar problems in parts of manuscript: eg. First page, Introduction; line 31 should be "to maximize safety"; line 33 should be "was shown" etc.
Answer : Dear reviewer thank you, we have made these corrections, and some others all through the text.
Reviewer 2
In this work the authors build on a the well established notion that conjugation of adjuvant and antigen results in stronger B- en T-cell responses. They present a strategy to couple larger (protein) antigens to PapMV nanoparticles; where previously only peptide were efficiently co-expressed. The authors should strengthen their manuscript by:
1) showing the physical characteristics of the newly formed nanoparticles
Answer : Dear reviewer, we have answered a similar question to reviewer 1 (question 2). In brief, we have added in figure S 1B the EM images of the free PapMV nano and PapMV nano coupled to either the NP-M2e and the GAG. We have also included figure S1A showing the dynamic ligth scattering of the free PapMV nano in comparison with those coupled to the NP-M2e or the GAG antigens. These data reveal that the coupling of the antigens to the PapMV nanoparticles did not affect the appearance of the nanoparticles. However, upon coupling, the size of the nanoparticles increased because f the coupling. We added the lanes 174-183 to explain these results.
2) fixing figure 1 (it contains strange symbols) ???? Incompatibility with computer software? It is fine here….
Answer : Dear reviewer, on ours ide the figure look OK? We will be carefull and manage this issue with the journals.
3) Supporting their claim that 15% of antigen was couple to the NPs (I can not pertain this from figure 1).
Answer : Dear reviewer, we have added information to better explain how the assessment of the coupling efficacy was performed in the material and methods at section 2.3 lane 105.
4) Discuss the large loss of antigen. Does this loss of antigen make this formulation pharmaceutically uninteresting?
Answer : Dear reviewer, I am not sure to understand your question. If you refer to the coupling reaction, I can tell you that there is no loss of antigen. If you look at figure 1B or D, you will see that most of the antigen NP or GAG present in the mixture (SDS PAGE lane 3) are efficiently attached to the PapMV CP (SDS PAGE lane 4). This is true for both antigens, GAG (Fig. 1B) and NP (Fig. 1D). The coupling efficacy of 15% refers to the pourcentage of PapMV CP that reacted with the antigen after the reaction with the sortase. I have added a sentence to address this point at lane 167 and to clarify the manuscript.
5) Describe the basis of Sortase A mediated conjugation. How do PapMV NPs accommodate this?
Answer : Dear reviewer, we have added a few sentences (lane 57) in the introduction and 3 references [14-16] that refers to litterature reviews that describe the mechanisms of the coupling by the SrtA.

Reviewer 2 Report
In this work the authors build on a the well established notion that conjugation of adjuvant and antigen results in stronger B- en T-cell responses. They present a strategy to couple larger (protein) antigens to PapMV nanoparticles; where previously only peptide were efficiently co-expressed. The authors should strengthen their manuscript by:
1) showing the physical characteristics of the newly formed nanoparticles
2) fixing figure 1 (it contains strange symbols)
3) Supporting their claim that 15% of antigen was couple to the NPs (I can not pertain this from figure 1).
4) Discuss the large loss of antigen. Does this loss of antigen make this formulation pharmaceutically uninteresting?
5) Describe the basis of Sortase A mediated conjugation. How do PapMV NPs accommodate this?
Author Response
Reviewer 2
In this work the authors build on a the well established notion that conjugation of adjuvant and antigen results in stronger B- en T-cell responses. They present a strategy to couple larger (protein) antigens to PapMV nanoparticles; where previously only peptide were efficiently co-expressed. The authors should strengthen their manuscript by:
1) showing the physical characteristics of the newly formed nanoparticles
Answer : Dear reviewer, we have answered a similar question to reviewer 1 (question 2). In brief, we have added in figure S 1B the EM images of the free PapMV nano and PapMV nano coupled to either the NP-M2e and the GAG. We have also included figure S1A showing the dynamic ligth scattering of the free PapMV nano in comparison with those coupled to the NP-M2e or the GAG antigens. These data reveal that the coupling of the antigens to the PapMV nanoparticles did not affect the appearance of the nanoparticles. However, upon coupling, the size of the nanoparticles increased because f the coupling. We added the lanes 174-183 to explain these results.
2) fixing figure 1 (it contains strange symbols) ???? Incompatibility with computer software? It is fine here….
Answer : Dear reviewer, on ours ide the figure look OK? We will be carefull and manage this issue with the journals.
3) Supporting their claim that 15% of antigen was couple to the NPs (I can not pertain this from figure 1).
Answer : Dear reviewer, we have added information to better explain how the assessment of the coupling efficacy was performed in the material and methods at section 2.3 lane 105.
4) Discuss the large loss of antigen. Does this loss of antigen make this formulation pharmaceutically uninteresting?
Answer : Dear reviewer, I am not sure to understand your question. If you refer to the coupling reaction, I can tell you that there is no loss of antigen. If you look at figure 1B or D, you will see that most of the antigen NP or GAG present in the mixture (SDS PAGE lane 3) are efficiently attached to the PapMV CP (SDS PAGE lane 4). This is true for both antigens, GAG (Fig. 1B) and NP (Fig. 1D). The coupling efficacy of 15% refers to the pourcentage of PapMV CP that reacted with the antigen after the reaction with the sortase. I have added a sentence to address this point at lane 167 and to clarify the manuscript.
5) Describe the basis of Sortase A mediated conjugation. How do PapMV NPs accommodate this?
Answer : Dear reviewer, we have added a few sentences (lane 57) in the introduction and 3 references [14-16] that refers to litterature reviews that describe the mechanisms of the coupling by the SrtA.
Reviewer 1
The paper is very interesting and I like the attention given to the immunological studies. Two comments I would like to make are:
(1) it is unclear from the Westerns on Figure 1B and D, lane 4, whether I am looking at an assembled nanoparticle here. The mw for the top band is close to 75 kD. So this is a fusion protein?
Answer : Dear reviewer thank you for your comment. To clarify the manuscript, we have modified the Figure 1B and D and identified with arrows the protein bands found on the gel and the western blots. Respectively, the PapMV CP monomer (Figure 1B and 1D), the NP-M2e (Figure 1B), the GAG (Figure 1D) and both fusion products (Figure 1B and 1D) were pointed with black arrows. We also identified the PapMV CP dimer (Figure 1B and D) that is the result of an incomplete denaturation of the PapMV nanoparticles on the SDS-PAGE.
(2) How do I know that any of this assembled into a nanoparticle and how long these particles are (compared to wt)?
Answer : Dear Reviewer thank you for your comment. We have added in supplementary figure S1A the dynamic ligth scattering (DLS) of the free PapMV nano and those coupled to the NP-M2e or the GAG antigens. At Figure S1B, we also showed the EM images of the free PapMV nano and coupled to either the NP-M2e and the GAG. These data reveal that the coupling of the antigens to the PapMV nanoparticles do not affect the appearance of the nanoparticles. However, the size of the nanoparticles has changed by the coupling. We have added lanes 174-184 to explain these observations. We prefer to keep this information in supplementary data because there are not critical for this manuscript.
(3) What is the cause of the difference in assembly ratios between GAG and NP?
Answer : Dear reviewer, thank you for your question. The coupling reactions for each antigen must be optimised. Those conditions will change depending of the nature of the antigen. Several factors can affect the conditions of coupling including the solubility of the antigen or the coupled antigen-nano complex. For example, the GAG antigen tends to aggregate at high concentration and also, as mentioned in the manuscript (lane 165-166), a higher density of coupling of GAG onto the PapMV nano induced formation of a precipitate that is unwanted for a vaccine preparation. Therefore, we were obliged to work with a ratio of PapMV CP/SrtA/antigen of 80 :4 :5 (µg/mL). This ratio contains less antigen in the coupling reaction than when the NP-M2e antigen is used because NP-M2e is more soluble allowing an optimal ratio of 80 :16 :16 (µg/mL). It also possible that the size of the antigen will have an impact on the total amount that we will be able to couple on the PapMV nano. As mentioned in the discussion (lane 281-283) that we have modified accordingly, the steric hindrance of large protein like the NP-M2e that are attached to the PapMV CP can mask an adjacent SrtA receptor motif and consequently affect the efficacy of coupling. Because peptides are smaller, it is possible to increase the coupling efficacy as compared to the large NP-M2e antigen and as previously reported (Thérien et al., 2017; ref. 7).
(4) Size and charge? From what I can see, I have no way of knowing that you have assembled nanoparticles except to look up reference 7.
Answer : (see response question 2, and the new Figure S1B)
(5) Any comments regarding the low efficacy of sortase mediated conjugation? Is it just that only a few CP fuse to antigen or is it because of steric hindrance that only 15% of antigen is present in nanoparticle?
Answer :(see answer question 3 and change to the discussion lane 281-283)
(6) Do you think it will be possible in the future to increase this percentage, and how would you propose to do that?
Answer : Dear reviewer, thank you for your comment. We are currently developing a second version of the vaccine platform where the antigen will be fused to the surface of the nanoparticle instead of the extremities as it is showed in this manuscript. It is anticipated that more SrtA receptor motif will be available at the surface of the nanoparticle, which will allow increasing potentially the % of efficacy of the coupling. However, these fusions will also mask the surface of the PapMV nanoparticle that migth be important for the recognition of the nanoparticle by the immune cells and for their targeting to the endosome where the antigen processing is performed. We are currently investigating this issues and hopefully we will be able by the end of this year to publish our results on a second generation of the PapMV vaccine platform.
(7) Some grammar problems in parts of manuscript: eg. First page, Introduction; line 31 should be "to maximize safety"; line 33 should be "was shown" etc.
Answer : Dear reviewer thank you, we have made these corrections, and some others all through the text.